# Investigating the Suitability of High Content Image Analysis as a Tool to Assess the Reversibility of Foamy Alveolar Macrophage Phenotypes In Vitro

**DOI:** 10.3390/pharmaceutics12030262

**Published:** 2020-03-13

**Authors:** Ewelina Hoffman, Darragh Murnane, Victoria Hutter

**Affiliations:** Department of Clinical and Pharmaceutical Sciences, School of Life and Medical Sciences, University of Hertfordshire, Hatfield, Hertfordshire AL10 9AB, UK; e.hoffman@herts.ac.uk (E.H.); d.murnane@herts.ac.uk (D.M.)

**Keywords:** foamy macrophages, high content analysis, phospholipidosis, apoptosis, macrophage morphology, vacuolation

## Abstract

Many potential inhaled medicines fail during development due to the induction of a highly vacuolated or “foamy” alveolar macrophage phenotype response in pre-clinical studies. There is limited understanding if this response to an inhaled stimulus is adverse or adaptive, and additionally if it is a transient or irreversible process. The aim of this study was to evaluate whether high content image analysis could distinguish between different drug-induced foamy macrophage phenotypes and to determine the extent of the reversibility of the foamy phenotypes by assessing morphological changes over time. Alveolar-like macrophages derived from the human monocyte cell line U937 were exposed for 24 h to compounds known to induce a foamy macrophage phenotype (amiodarone, staurosporine) and control compounds that are not known to cause a foamy macrophage phenotype in vitro (fluticasone and salbutamol). Following drug stimulation, the cells were rested in drug-free media for the subsequent 24 or 48 h. Cell morphometric parameters (cellular and nuclear area, vacuoles numbers and size) and phospholipid content were determined using high content image analysis. The foamy macrophage recovery was dependent on the mechanism of action of the inducer compound. Amiodarone toxicity was associated with phospholipid accumulation and morphometric changes were reversed when the stimulus was removed from culture environment. Conversely cells were unable to recover from exposure to staurosporine which initiates the apoptosis pathway. This study shows that high content analysis can discriminate between different phenotypes of foamy macrophages and may contribute to better decision making in the process of new drug development.

## 1. Introduction

Alveolar macrophages reside on the luminal surface of the alveolar space and are a normal feature of the healthy lung [1]. They are the first immune cells in the airways to encounter incoming pathogens, pollutants, or particles and play a crucial role in the initiation and resolution of immune responses in the lung [1,2]. Alveolar macrophage responses are commonly observed in assessing the safety of novel inhaled medicines during in vivo pre-clinical studies in rats [3]. Currently alveolar macrophage responses in the lungs reported in pre-clinical studies are largely limited to observations of an increase in the number of immune cells and the presence of cells with a highly vacuolated or “foamy” morphology [3]. Novel inhaled medicines are often withdrawn from further development due to concerns regarding the potential toxicity manifested by the presence of these foamy macrophage observations in pre-clinical animal studies [3]. These “foamy” macrophage (FM) responses can be accompanied by additional abnormal lung tissue changes such as neutrophil infiltration or lymphocyte degeneration [4]. The pathophysiology of foamy macrophage development is not explicit. The highly vacuolated phenotype may be induced by a variety of cellular mechanisms including phagocytosis of poorly soluble drug particles, stimulation of excess lung surfactant, impaired metabolism of intracellular phospholipids, phagocytosis of surfactant phospholipids, apoptosis, and autophagy [3,4,5]. It is not clear whether increased alveolar macrophage vacuolation should be classified as a permanent an adverse or adaptive response [3,4]. The majority of in vitro toxicity testing is focused on the acute effects of compounds at high concentrations [6]. However, this strategy may not reflect cell responses during long-term exposures and low-dose toxicant effects. To better understand the foamy alveolar macrophage phenomenon, it is necessary to study the timeframe over which increased vacuolation occurs and whether there are morphological and/or biochemical markers, or thresholds of these markers which indicate if the foamy macrophage phenotype is transient and reversible or a permanent and irreversible process.

Pre-clinical safety of new inhaled medicines is predominantly assessed in vivo using rats. These studies typically range from 30 to 90 day studies sacrificing a minimum of six rats per time point over a range of concentrations, which in total require approximately 180 rats in a single drug efficacy study [7]. Endpoints to assess inflammatory changes rely on histopathology examination of lung slices which are assigned a qualitative description of inflammation or pathological changes. There is no universally standardized procedure for these in vivo assessments, and they are time and resource consuming [3,5]. Additionally, immune responses in the lungs of rats are known not to correlate well with that of healthy human lungs [8]. Rats are obligate nose-breathers with marked differences in their lung anatomy, physiology, and biological responses and as a result are more sensitive to inflammatory responses compared with humans [3,9]. Studies using alveolar macrophages harvested from animals are challenging due to the inaccessibility of cells. Additionally, the alveolar macrophage population in rat bronchoalveolar fluid is very small, and the lavage procedure may significantly affect macrophage health and phenotype, resulting in an insufficiently healthy cell population for further tests [10]. Therefore, human in vitro alveolar-like macrophage models may offer an advantage by generating quantitative data allowing for detailed assessment of macrophage health and function. Furthermore, such models allow for longitudinal assessment of morphology, functionality, and activation state of different FM phenotypes.

Until recently, cell vacuolation in foamy alveolar macrophage responses was predominantly assessed visually using light microscopy to image cells from bronchoalveolar lavage (BAL). Researchers could only categorize foamy cells into two arbitrary groups: either finely or coarsely vacuolated [3]. We have developed high-throughput methodology allowing to accurate quantify a range of vacuole characteristics including the number of vacuoles per cell, their size, and the proportion of the cell they occupy in relation to other cellular features in a 2D cross-section of the cell [11]. Quantitative characterization of cell vacuolation patterns alongside other markers of cell health and functionality may help to categorize foamy alveolar macrophage responses in more detail distinguishing between different FM phenotypes.

The aim of this study was to determine if high content image analysis could be used to categorize morphological responses of alveolar macrophages to different stimuli and follow these changes over time to assess the extent of reversibility. Four compounds were selected to represent different pathways of foamy macrophage induction and control compounds: (i) amiodarone, a cationic amphiphilic drug known to induce a FM phenotype as a result of phospholipidosis in vitro and in vivo [12]; (ii) staurosporine, an established inducer of apoptosis which creates a foamy cell phenotype during early stage apoptosis by increased vacuolation and membrane blebbing [13,14]; (iii) fluticasone, a marketed corticosteroid used for inhalation therapies [15]; (iv) salbutamol, a marketed short-acting, selective β_2_-adrenergic agonist used in treatment of asthma and chronic obstructive pulmonary disease (COPD) [16]. Previous studies have established that amiodarone and staurosporine induce defined changes in the vacuolation pattern of macrophages in vitro [11]. In contrast, salbutamol and fluticasone, both marketed inhaled active pharmaceutical ingredients (APIs) are not known to cause a foamy alveolar macrophage phenotype in vitro and are used as a baseline control for the purposes of this study [17].

## 2. Materials and Methods

### 2.1. Cell Culture

U937 human monocyte cells were cultured in RPMI-1640 cell culture medium (Sigma Aldrich, Dorset, UK) supplemented with 10% *v*/*v* fetal bovine serum (Sigma Aldrich, Dorset, UK), 1% *v*/*v*
l-glutamine (Sigma Aldrich, Dorset, UK) and 1% *v*/*v* penicillin/streptomycin (Sigma Aldrich, Dorset, UK) and incubated in a humidified atmosphere at 37 °C and 5% *v*/*v* CO_2_. For experiments, cells were used between passage numbers of 2 and 20 and were seeded onto 96-well plates at density of 25,000 cells/well. For differentiation into alveolar-like macrophages, cells were incubated in complete cell culture medium containing 100 nM phorbol myristate acetate (PMA) (Sigma Aldrich, Dorset, UK) following methodology described previously [18,19,20]. The differentiation process was followed by 24 h resting period in PMA-free complete cell culture media.

PMA-differentiated alveolar-like macrophages were exposed to 0.1–50 µM solutions of either amiodarone, salbutamol, fluticasone, or staurosporine in complete cell culture media. The concentrations for this study were selected from preliminary studies which indicated morphological changes were induced (Appendix B, Figure A2) without affecting cell viability (Appendix A, Figure A1). All compounds were purchased from Sigma Aldrich (Sigma Aldrich, Dorset, UK). After 24 h of active pharmaceutical ingredient (API) exposure, the cell culture media was replaced with fresh media without stimuli and cells were incubated at 37 °C, 5 % *v*/*v* CO_2_ for a further 24 or 48 h to assess their recovery.

### 2.2. Assessment of Cell Vacuolation and Lipid Profiles

Fluorescence staining and imaging were performed as described previously by Hoffman et al. [11] to determine the cellular lipid content and to assess macrophage morphology. Briefly, macrophages were incubated simultaneously with the tested compound and HCS LipidTox Phospholipid Red dye (Invitrogen, Renfrewshire, UK) diluted 1:1000 (according to the manufacturer’s protocol) for 24 h. After the desired compound exposure time, the cells were fixed with 3.7% *w*/*v* paraformaldehyde containing Hoechst 33342 (10 µg/mL) for 20 min, followed by one washing step with 100 µL phosphate buffered saline (PBS). The cells were then incubated with Cell Mask Deep Red (Invitrogen, Renfrewshire, UK) diluted 1:1000 (according to the manufacturer’s protocol) for 30 min at room temperature for morphometric characterization. The cells from the assay were stored in the dark at 4 °C before sample acquisition. Images were captured using the InCell Analyser 6000 (GE Healthcare, Little Chalfont, Bucks, UK) with a 40× objective in standard 2D imaging mode with an exposure time of 0.1 s.

### 2.3. Quantitative High Content Analysis

Image analysis was performed using In Cell Developer Toolbox v 1.9.2, level 3 analysis (GE Healthcare, Little Chalfont, Bucks, UK) using a method previously published by Hoffman et al. [11]. In brief, the cell nuclear dye Hoechst 33342 was used to identify nucleated cells, while Cell Mask Deep Red staining highlighted cytoplasm and allowed quantification of vacuoles based on negative staining. Vacuoles within cells were identified based on negative staining and quantified using a grey scale threshold of 19,000 to minimize any background noise and to include only true vacuole features [11]. The intracellular accumulation of phospholipids was detected and quantified by incubation with LipidTox Phospholipid Red and reported as fluorescence intensity values [11]. Each well of the 96-well plate was imaged using 15 fields representing in total between 500 and 1500 cells per well. Quantitative measurements for each cell were generated from the image analysis for cell area, vacuole number per cell, vacuole area per cell, and phospholipid content per cell.

### 2.4. Statistical Analysis

All experiments were repeated independently three times (on three different cell passage numbers). Two-way ANOVA analysis with Bonferroni multiple comparison post-hoc tests were used to assess the statistical significance between the different cell models. Statistical significance was evaluated at a 95% confidence level (*p* < 0.05). All statistical tests were performed using Graphpad Instat^®^ version 3.06.

## 3. Results

### 3.1. Characterization of PMA-Differentiated U937 Cells

Cell data for differentiated U937 cells after 24 and 72 h was assessed to establish human alveolar-like macrophage morphological characteristic in vitro over time in culture (Figure 1). U937 cell area ranged from 511 to 686 µm^2^. The untreated macrophage-like cells demonstrated similar vacuole profiles at both time points assessed. The average number of vacuoles per cells was 72 (±9.9) after 24 h and 81 (±5) when cells were cultured up to 72 h, while an average of 29% and 27% of cell area was occupied by vacuoles after 24 and 72 h incubation, respectively. An average individual vacuole occupied 4.4% (±2.2) of the cell area after 24 h incubation and 4.8% (±2.2) after 72 h. No significant differences (*p* > 0.05) were observed for the cellular area and the vacuolation parameters assessed, confirming the suitability of the model to assess changes in vacuole morphology over time.

### 3.2. Phospholipidosis Phenotype

The cationic amphiphilic API, amiodarone was used as an established inducer of phospholipidosis [12]. Cells were challenged with the drug for 24 h in concentrations which induced cellular phenotype changes but were not toxic to the cells (Appendix A, Figure A1). Macrophage-like cells exposed to amiodarone did not display a significant difference (*p* > 0.05) in cellular area nor the area of the cell occupied by vacuoles (Figure 2a,c). However, significant changes (*p* < 0.001) in the number of vacuoles were observed compared with untreated samples (Figure 2b). Cells challenged for 24 h with 5 and 10 µM amiodarone contained 1.8 (*p* < 0.001) and 2.6 (*p* < 0.001) times fewer vacuoles, respectively, when compared with untreated cells. To investigate cellular ability to recover after the initial short-term amiodarone exposure (24 h), the drug was washed out and the cells were further incubated in drug-free media for 24 and 48 h. The number of vacuoles returned to values of untreated cells after 48 h resting time. At the same time, the area of the cell occupied by vacuoles was unchanged, which suggests that vacuoles were smaller in size after removal of the stimulus. After 48 h in amiodarone-free media macrophage morphology resembled the vacuolation pattern of 24 h untreated cells. Despite observed signs of morphometric recovery, the cells still accumulated phospholipids. Cells challenged with 10 µM amiodarone for 24 h followed by a 48 h API-free incubation period displayed 2.9 (*p* < 0.001) times higher phospholipid content when compared with untreated cells (Figure 2d).

### 3.3. Pro-Apoptotic Phenotype

Staurosporine, a known pro-apoptotic agent, induced the process of cell death, which was reflected in the cell morphometric parameters. U937 cells treated with staurosporine displayed significantly reduced (*p* < 0.001) cellular area to 12% after 24 h incubation (Figure 3a). The cells significantly reduced in size (*p* < 0.001), which is a characteristic feature of apoptosis. Staurosporine treatment also resulted in a significant decrease (*p* < 0.001) in the number of vacuoles per cell from 72 (±10) (untreated cells) to nine (±2) after 24 h exposure. These observations were not reversed after the removal of staurosporine from the culture conditions and the number of vacuoles per cell remained 9 (±11) (Figure 3b). Similarly, the cell area occupied by vacuoles significantly decreased (*p* < 0.001) after 24 h from 29% (±3.2) (untreated cells) to 9.8% (±1.4) and remained unchanged after removal of the stimulus from the culture media (Figure 3c).

### 3.4. Corticosteroid Phenotype

Fluticasone propionate is a glucocorticosteroid API used in treatment of inflammation in various diseases, including such lung conditions as asthma or COPD [15]. In this study, it did not induce significant changes (*p* > 0.05) in U937-derived macrophage morphology (Figure 4a–c). However, significant changes were noted in cellular phospholipid content (Figure 4d). Initially, after 24 h, fluticasone did not affect phospholipid content. However, elevated phospholipid content was noted following drug wash-out; 2.6 (*p* < 0.001) and 1.8 (*p* < 0.01) times higher after 24 and 48 h recovery, respectively (Figure 4d).

### 3.5. β_2_-Agonists Phenotype

Salbutamol is a β_2_-agonist used as bronchodilator to relieve symptoms of asthma and COPD [16]. Similarly, to fluticasone, salbutamol did not affect cell morphology, nor the cellular phospholipid content (Figure 5).

## 4. Discussion

Cellular recovery and resilience to toxic insult has not been studied to a great extent in in vitro toxicology testing. The concept of cellular resilience has been described by Smironova et al. [21], defined as the cell’s ability to cope with any perturbation or environmental changes and recover. The authors hypothesized that cells can overcome low-dose toxic effects and become either resilient or more susceptible to subsequent exposure [21]. In this study human monocyte-derived alveolar-like macrophages were used to determine whether effects of short-term (24 h) drug exposure were permanent or reversible.

The term “foamy macrophage” describes lung macrophages with highly granular or vacuolated cytoplasmic appearance when viewed under a light microscope. The term “vacuoles” used in this study refers to morphological feature of a cell and includes endosomes, lysosomes, as well as lipid droplets. These vacuolated macrophages are typically described also as enlarged when compared to non-vacuolated cells [3]. The majority of studies reporting alveolar macrophage morphology describe qualitative observations. Usually, the morphological changes resulting from treatment with various compounds are so pronounced that quantification may seem unnecessary. For example, the loss of cell volume or cell shrinkage, alongside nuclear condensation, are the predominant signatures of apoptosis [22]. Such apoptotic characteristics can be observed using light or fluorescent microscopy [23]. However, practice is required to accurately identify the different types of deteriorated cells. Furthermore, some morphological changes may be difficult to interpret and determining the specific process responsible for altered morphology is challenging using the current, standard techniques [3,4]. Currently, FM appearance is assessed only by light microscopy as one of the endpoints of histopathological studies [4]. Moreover, the macrophage vacuolation patterns of primary cells obtained from tested animals have not been described in detail before. The only differentiation that has been made to-date was for finely and coarsely vacuolated cells and was based on light microscopy images [3].

The high content image analysis technique reported in this study demonstrates it is possible to measure detailed morphological characteristics including cellular area, report the number of vacuoles, and quantify the area of the cell occupied by vacuoles rapidly. Furthermore, such parameters can be assessed over a time course, which may further help to determine if a particular phenotype is adaptive or adverse. Quantitative determination of cellular morphology, including cell vacuolation pattern, employed in the early phase of drug development may help to identify FM-inducing compounds early so that the number of such compounds entering in vivo safety trials would be reduced. The majority of in vitro methodologies currently employed for inhaled safety assessment use short-term, single high-dose exposures and only consider the acute cellular response [24,25]. Whilst these test conditions may capture early cellular responses the significance of these effects on medium-long term cell health and function are not considered. As a result, consideration of the responses being reversible and a normal adaptive response to a given stimuli or chronic consequence of exposure are overlooked. Employment of more accurate and sensitive techniques, such as high content image analysis, in early in vitro studies would allow for longitudinal studies of a xenobiotic (e.g., API) toxicity, as well as potential cellular recovery to better inform decision making.

High content analysis employed in this study permitted a more detailed classification of FM responses than light microscopy. Whilst images taken using light microscopy permit classification of “foamy” vs. “non-foamy” cells, the current study demonstrates that detailed morphology and vacuolation parameters can be acquired using high content image analysis techniques to elucidate the mechanism of cellular response to a drug challenge (phospholipidosis vs. apoptotic pattern). Moreover, changes can be tracked over time which gives an opportunity to observe how cells respond to stimuli and whether any changes in cell morphology can be potentially reversible.

### 4.1. Phospholipidosis Phenotype Was Not Destructive for Macrophages

The induction of phospholipidosis by amiodarone is well-established [26] and was also confirmed in the current study. The postulated mechanism of FM development accompanying phospholipidosis is based on the ability of the amiodarone molecule to bind to phospholipids and prevent enzyme-mediated metabolism, as well as direct inhibition of phospholipases themselves [27,28]. In this study, cellular phospholipid content was measured using a standard indirect method employing a fluorescent dye (HCS LipidTox Phospholipid Red). The recent publication by Patel et al. confirmed that this technique reflects the true phospholipid content in a cell as quantified using mass spectrometry imaging for cells exposed to amiodarone, and the binding of the fluorescent dye was not directly affected by presence of the drug [26]. Phospholipid accumulation was observed at both concentrations tested (5 and 10 µM) with no detrimental impact on cell health (Appendix A, Figure A1). API-induced phospholipidosis is a condition defined by the appearance of intracellular accumulation of phospholipids and lamellar bodies [29]. There are studies showing that alveolar macrophages have the ability to scavenge lamellar bodies and become foamy when completely packed by lipids [27]. Phospholipids are accumulated in a time- and dose-dependent manner. The biological consequence of phospholipid accumulation in the lungs has not been evaluated but it is usually linked to cellular toxicity. However, a review by Anderson and Barlock suggested that phospholipidosis might be a part of a detoxification mechanism to protect cells from xenobiotics [29]. Effects of amiodarone-induced phospholipidosis on alveolar macrophages has been reported and Reasor et al. [30] concluded that amiodarone does not impair pulmonary host defense processes. Moreover, it may be associated with augmentation of some activities [30]. In this study, macrophages exposed to amiodarone showed increased vacuolation, which was reflected in fewer but larger vacuoles after an initial 24 h exposure. The vacuolation pattern resumed that of untreated cells after the recovery time. Despite this morphometric reversibility, cells still accumulated phospholipids. Similar findings have been reported previously in rat alveolar macrophages exposed to another cationic, amphiphilic API, chlorphentermine [31]. Despite biochemical recovery of alveolar macrophages after 12 days in vitro culture, a moderate level of lamellar bodies was still present in macrophages [31]. Prolonged phospholipid accumulation, even after an API wash out, may be due to potential intracellular API retention and further release from the multiwell plate itself. Kramer et al. [32] studied the distribution of compounds in the in vitro system, measured by the concentration of APIs in cells, multiwell plate plastic, and medium. Results indicated that lipophilic APIs bind significantly to plastic labware [32]. Additionally, some compounds, including amiodarone, accumulate significantly in the cell over time [32,33]. It has been speculated that highly lipophilic amiodarone targets and accumulates in cellular phospholipid bilayer and interferes with the fatty acyl alignment [34]. Moreover, some metabolites of amiodarone have been found to contribute towards alveolar macrophage toxicity [33].

### 4.2. Apoptotic Phenotype Was Not Reversible

Staurosporine is a very potent compound used in many studies as positive control for the process of programmed cell death (apoptosis) [35]. Early during the initiation of apoptosis, cells start to show protrusions of the plasma membrane commonly referred to as blebs [36]. The cellular area is decreased, and finally the blebs separate forming apoptotic bodies densely packed with cellular organelles and nuclear fragments [37]. The shrinkage of the cells, blebbing, and forming of apoptotic bodies can be observed using light microscopy [22,36]. In this study, cells exposed to staurosporine were smaller in size with reduced numbers of vacuoles and a smaller cell area occupied by vacuoles. These morphological changes are in agreement with reported descriptions of apoptotic cell morphology where dense cytoplasm and tightly packed organelles are also observed [37]. These changes were observed regardless of time point, or cell acute or recovery status. It has been reported that apoptosis can be reversed depending on the pathway which it has been activated (e.g., early stages of p53-induced apoptosis are reversible [38]) or how early a stimuli has been removed [39]. There is evidence that dying cancer cells can regain their normal morphology and proliferate after removal of apoptotic inducers [39]. In this study, staurosporine-induced apoptosis resulted in irreversible changes. Staurosporine acts through activation of caspase cascade pathway [40,41], which when activated is considered unstoppable [37]. Further studies are therefore required to identify a “cellular checkpoint” that, once crossed, makes changes persistent.

### 4.3. Corticosteroid and β_2_-Agonist Phenotypes Resolve over Time

Fluticasone propionate (glucocorticosteroid) and salbutamol hemisulfate (β_2_-agonist) are both routinely used for the treatment of lung disorders including COPD and asthma [42]. Fluticasone has anti-inflammatory properties [43], while salbutamol is used as bronchodilator [44]. Both APIs have been available on the market for more than 30 years and can be co-administered together or used as a single-agent treatment [45].

It has been demonstrated that the FM responses associated with the in vivo administration of corticosteroids to animals were located at the bronchoalveolar junctions and are linked with accumulation of various lipids [4]. The in vitro results of fluticasone-induced macrophage response are partly in agreement with in vivo findings. Although morphometric changes were not observed and the presence of foamy macrophages was not confirmed in the U937-derived in vitro macrophage model, it was shown that 50 µM concentration of fluticasone increased phospholipid accumulation following 24 + 48 h API incubation. Lewis et al. argued that corticosteroid-induced FM responses were not progressive in animal models [4]. The results of the current in vitro study also suggest that corticosteroid macrophage phenotype may be resolved over time, since the accumulation of phospholipids was significantly (*p* < 0.001) reduced the longer the stimulus was removed.

Similarly, the in vivo administration of β_2_-agonists may induce FM response located at the bronchoalveolar junctions and is linked with accumulation of surfactant phospholipids [4]. Salbutamol treatment in vitro did not confirm these findings, as no FM morphology was observed, nor significant changes in phospholipids accumulation.

Normal physiological changes to the morphology of alveolar macrophages is anticipated as a response to non-hazardous and toxic inhaled substances. This study has demonstrated that minor changes in macrophage phenotypes can be induced from exposure to APIs which have been available on the market for decades (e.g., fluticasone, salbutamol) and have considerable safety data in humans. High content image analysis has allowed these cellular responses to be quantified, showing marked differences between cell responses to different stimuli and demonstrate the ability to monitor these changes over time. This technique may make it possible to ascertain a threshold of where a cellular response changes from a normal adaptive response to an adverse event.

## 5. Conclusions

This work provides a useful contribution to better understanding the foamy macrophage phenotype associated with API stimulation. Potential differences in alveolar macrophage responses to several compounds were assessed and how these changes may resolve over time. Collectively, our results indicate that FM phenotypes may resolve differentially, leading the cell either to apoptosis or adaptive changes. The fate of the affected cell is determined by its mechanism of action. Our study provides one of the first insights into the reversibility of FM changes which in turn may aid a more relevant FM response classification and enhance decision making in the process of new inhaled medicines development.

Further research is needed to better understand whether resilience mechanisms are beneficial or detrimental to cells in the long-term. Prior stimulation can lead to different responses to subsequent stimuli. Macrophages after recovery may be either more resistant or susceptible to further exposure (via activation of cell survival/death pathways, changes in gene expression, or epigenetic modifications). Furthermore, permanent activation or inhibition of specific pathways may contribute to disease pathology. Previous inflammatory episodes may critically modify function and reactivity of cells. Further studies will investigate if resting alveolar macrophages retain the same responsiveness to stimuli as non-stimulated previously cells.

## Figures and Tables

**Figure 1 pharmaceutics-12-00262-f001:**
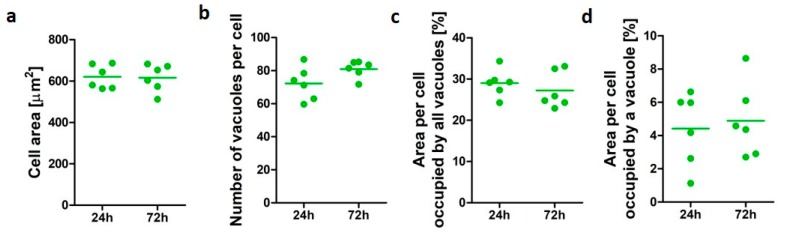
Morphometric characterization of untreated alveolar-like macrophages (U937). Cell area (**a**), number of vacuoles per cell (**b**), area per cell occupied by all vacuoles (**c**), and average area per cell occupied by a vacuole (**d**) after culture in complete cell culture medium for 24 and 72 h. All results are presented as mean of *n* = 6 ± SEM of three independent experiments.

**Figure 2 pharmaceutics-12-00262-f002:**
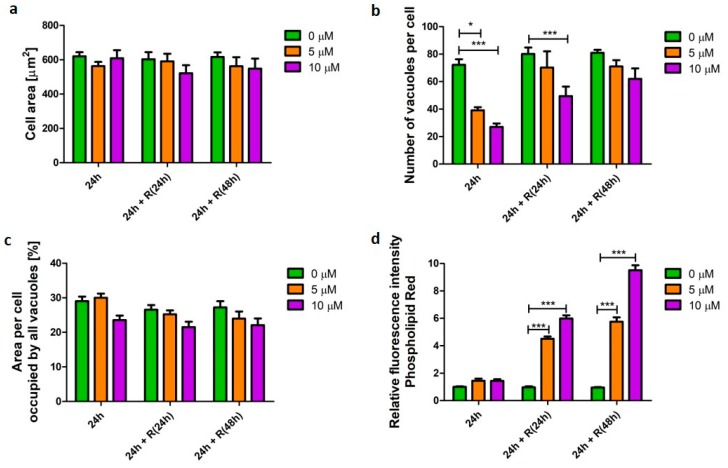
Characteristics of alveolar-like macrophages (U937) with an amiodarone-induced phospholipidosis phenotype. Cells were exposed to phospholipidosis-inducer amiodarone at concentrations of 5 (orange bars) and 10 µM (purple bars) for 24 h followed by recovery (R) in the absence of the stimulus for 24 and 48 h. Cell area (**a**), number of vacuoles per cell (**b**), area per cell occupied by all vacuoles (**c**), and relative change of cellular phospholipid content (**d**) were quantified and presented as mean ± SEM of three independent experiments. * indicates *p* < 0.05; *** indicate *p* < 0.001.

**Figure 3 pharmaceutics-12-00262-f003:**
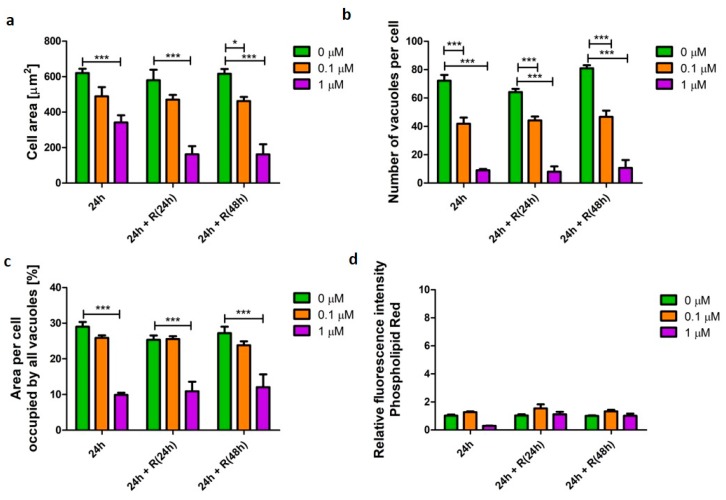
Characteristics of alveolar-like macrophages (U937) with a staurosporine-induced pro-apoptotic phenotype. Cells were exposed to apoptosis inducer staurosporine at concentrations of 0.1 (orange bars) and 1 µM (purple bars) for 24 h followed by recovery (R) in the absence of the stimulus for 24 and 48 h. Cell area (**a**), number of vacuoles per cell (**b**), area per cell occupied by all vacuoles (**c**), and relative change of phospholipid cellular content (**d**) were quantified and presented as mean ± SEM of three independent experiments. *** indicates *p* < 0.05; ***** indicate *p* < 0.001.

**Figure 4 pharmaceutics-12-00262-f004:**
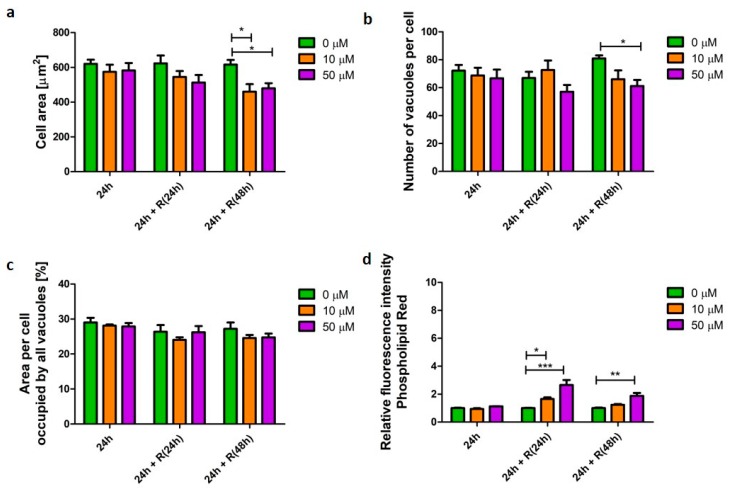
Characteristics of macrophages (U937) with a fluticasone-induced corticosteroid phenotype. Cells were exposed to a steroid (fluticasone propionate) at concentrations of 10 (orange bars) and 50 µM (purple bars) for 24 h followed by recovery (R) in the absence of the stimulus for 24 and 48 h. Cell area (**a**), number of vacuoles per cell (**b**), area per cell occupied by all vacuoles (**c**), and relative change in cellular phospholipid content (**d**) were quantified and presented as mean ± SEM of three independent experiments. * indicates *p* < 0.05; ** indicate *p* < 0.01; *** indicate *p* < 0.001.

**Figure 5 pharmaceutics-12-00262-f005:**
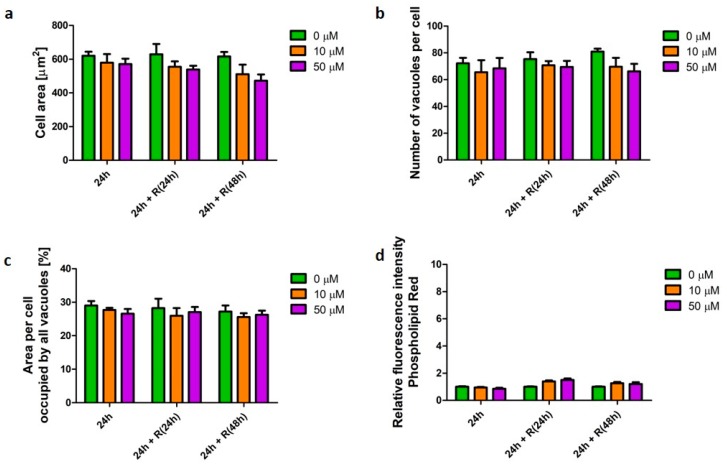
Characteristics of macrophages (U937) with a salbutamol-induced β-agonist phenotype. Cells were exposed to salbutamol at concentrations of 10 (orange bars) and 50 µM (purple bars) for 24 h followed by recovery (R) in the absence of the stimulus for 24 and 48 h. Cell area (**a**), number of vacuoles per cell (**b**), area per cell occupied by all vacuoles (**c**), and relative change of cellular phospholipid content (**d**) were quantified and presented as mean ± SEM of three independent experiments.

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
