# Peer review of "Investigating the Suitability of High Content Image Analysis as a Tool to Assess the Reversibility of Foamy Alveolar Macrophage Phenotypes In Vitro"

_pharmaceutics, 2020, doi:10.3390/pharmaceutics12030262_

Round 1
Reviewer 1 Report
In the present study, Hoffman and colleagues attempted to determine whether high content image analysis could distinguish different drug-induced foamy macrophage phenotypes and to determine the extent of the reversibility of the foamy phenotypes. Using the human monocyte cell line, U937, the authors investigated the impact of four different drugs on cell morphometric parameters and phospholipid content using high content image analysis. The authors confirmed the activity of amiodarone on the phospholipid content and the pro-apoptotic activity of staurosporine. Further, they found that treatment with fluticasone slightly reduced the cell area and number of vacuoles per cell. The treatment of salbutamol showed no impact on cell morphology and cellular phospholipid content.
However, this study is more like a preliminary study for screening several compounds. One of the aims of this study was to determine the benefits of high content image analysis. However, the authors didn’t include any images in the manuscript. The results were hard to support the conclusions. There are numerous major and minor issues throughout the whole manuscript.
Major compulsory revisions
Sentences through the whole manuscript are loosely connected, and the writing lacks proper transition and flow which not only makes it difficult to understand but also confuse the reader. Title: It doesn’t fully cover the content in the manuscript. And also, choose either “in U937 cells” or “in vitro”. Introduction: It’s better to explain how phospholipids are involved in the pathophysiology of “foamy” macrophage. The rationale for choosing this model is hard to convince the reader. Materials and Methods: Provide the concentration of PMA used for treating the U937 cells. The information in the Line 111-116 is very confusing. Results: The “3.1 Characterization of PMA-differentiated …” could be included as supplemental figures. Provide a rationale for choosing different concentrations of different drugs.Minor essential revisions
1) “API” in Line 105 should be spelled out.
2) In Line 201-205, “Figure 5” should be “Figure 4”.
3) The “Figure 6” in Line 216 should be “Figure 5”.
4) Miss references for the statements starting in Line 302, Line 321.
Author Response
Major compulsory revisions
Point 1: Sentences through the whole manuscript are loosely connected, and the writing lacks proper transition and flow which not only makes it difficult to understand but also confuse the reader.
Response 1: The authors disagree with the point. The manuscript was written and edited by native English speakers at Associate Professor and Professorial level with a track record of dozens of publications in the field. The sentence structure is sound and sentences link clearly to form a well-constructed narrative.
Point 2: Title: It doesn’t fully cover the content in the manuscript. And also, choose either “in U937 cells” or “in vitro”.
Response 2: Whilst the original title reflected the aim of presented study, for improved clarity and increased specificity the authors have added the technique employed in this research. The phrase “in U937 cells” has been removed.
Point 3: Introduction: It’s better to explain how phospholipids are involved in the pathophysiology of “foamy” macrophage.
Response 3: The pathophysiology of a ‘foamy’ macrophage is not explicit – it is a general term used in the field (see ref 3 and 4) used to describe a macrophage with a highly vacuolated appearance or ‘foamy’ appearance under light microscopy. One established phenotype which displays a ‘foamy’ response is phospholipidosis. For better clarity a postulated mechanism of foamy macrophage induction by amiodarone has been added in Lines 291-294.
Point 4: The rationale for choosing this model is hard to convince the reader.
Response 4: The rationale of choosing human alveolar-like macrophages differentiated from U937 has been explained and justified in paragraph 2 of Introduction: Lines 65 – 75. Additionally, the appropriateness of this model and similar models has been established through several publications (e.g. refs 11, 17)
Point 5: Materials and Methods: Provide the concentration of PMA used for treating the U937 cells.
Response 5: The concentration of PMA (100 nM) has been added to Section 2.1 of Material and Methods - Line 106.
Point 6: The information in the Line 111-116 is very confusing.
Response 6: The information has been paraphrased for better clarity in Lines 120-124.
Point 7: Results: The “3.1 Characterization of PMA-differentiated …” could be included as supplemental figures.
Response 7: The authors believe the information in section 3.1 is essential for the reader identify the baseline for alveolar-like macrophage morphology over time especially in the context of the reversibility of morphological features which is one of the key questions in this article. Figure 1 presents a baseline of healthy macrophages to which all treatments were compared.
Point 8: Provide a rationale for choosing different concentrations of different drugs.
Response 8: Drug concentrations chosen for this study have not affected cell viability after 24 hours drug exposure, but they have been shown to induce morphological changes to macrophages. This information was added in Lines 110 – 113.
Minor essential revisions
Point 1: “API” in Line 105 should be spelled out
Response 1: “API” has been extended to “active pharmaceutical ingredient” (Line 114).
Point 2: In Line 201-205, “Figure 5” should be “Figure 4”
Response 2: Amended as advised.
Point 3: The “Figure 6” in Line 216 should be “Figure 5”
Response 3: Amended as advised.
Point 4: Miss references for the statements starting in Line 302, Line 321.
Response 4: Missing references have been added: Line 332 - reference number 37; Line 349 – references number 43 and 44; Line 351 – reference number 45.
Reviewer 2 Report
Please, see attached file

Author Response
Point 1: A first primary question is how much the cells that have been used to carry these experiments are really alveolar macrophages. The true alveolar cells have a very particular phenotype and a singular response as a consequence to the exposure to external agents. It would have been particularly useful to validate some of the observations in true alveolar macrophages isolated from bronchoalveolar lavage.
Response 1: The presented study is not attempting to mimic in vivo situation. The aim of this work was to evaluate whether high throughput image analysis (HCA) can be employed to look at longitudinal morphometric changes in macrophages. The authors make it clear throughout the article that the cells are alveolar-like macrophages. Whilst it is appreciated that the cell source is not directly from an alveolar macrophage lineage there are no human alveolar macrophage cell lines available. The U937 cells in this study have been previously optimised to represent that of alveolar macrophage phenotype, morphology and function as closely as possible.
Point 2: Other important question is that the behaviour of alveolar macrophages at their real
pulmonary context is modulated by the presence of lung surfactant. The presence of this additional element would be important when assessing the response and durability of the effects of drugs on the macrophage phenotype. The authors could have tested the effect of the presence of certain amount of any of the available surfactants on the cells, or at least including this element into the discussion.
Response 2: The impact of surfactant on macrophage responses is an interesting point for further investigation. Similarly, the interplay among various type of cells in alveolar microenvironment e.g. epithelial cells type I and type II. However, these questions are outside the scope of the paper. Rather than mimicking exactly the in vivo situation, the paper seeks to present a snapshot of macrophage morphometric responses, as well as to evaluate if HCA is a powerful technique to analyse these responses. Although for physiological conditions, macrophages in the alveolar spaces are subject to regulation by surfactant or other cell types, the action of individual drugs on isolated alveolar-like macrophage phenotype is important in postulating their role in inhaled safety assessment.
Point 3: It is absolutely crucial to include in the paper some illustrative pictures of the different cellular phenotypes as they evolve.
Response 3: Representative images have been added as Appendix B, Figure B1.
Point 4: Related with the previous point, a major concern when obtaining quantitative data
from microscopy analysis is how much numbers are biased by the quality of the microscopy. For instance, how the potential problems with out of focus vacuoles have been considered when determining data as the total number of vacuoles and the total “area” occupied by vacuolar structures per cell. It seems that the proposed approximation is assuming that the images as obtained in 2D optical slices are representative enough of the full cellular volume. Please, add technical details at the methods to explain how to cope with 2D “projections”. The question could be particularly problematic when the number of vacuoles changes (with or without changes in the total vacuolar area), because coalescence or fragmentation of vacuoles could be particularly masked by limited optical definition.
Response 4: The 2D image taken is not intended to represent the 3D volume of the cell. The cell areas and data per cell referred to in the results is related to the 2D slices taken as would be standard practise for any image analysis technique. The IN-Cell Developer Toolbox was used to analyse vacuolation within the cell. Thresholds were set and optimised to minimalize ‘false’ images of vacuolation. Additionally, this technique has assessed 500-1500 cells per well and as such occasional anomalies do not skew the data. For clarity as well as referring the reader to the previously published methods (Ref. 11 – Line 140) a detailed sentence regarding vacuole identification and thresholding has been added for clarity (Lines 138 - 140).
Point 5: Please, clarify, in the methods and the legends of the corresponding figures, how many samples and repetitions have been carried out. For instance, data in Figure 1 (6 data points on area, number of vacuoles, vacuolar area per cell, area per vacuole per cell) come from “n=6 wells from three independent experiments”. Need to explain unambiguously how each data point in the graph has been calculated. The same for the error bars in figures 2, 3, 4, 5, and A1.
Response 5: All experiments were repeated independently three times. This information has been added in Section 2.4 of Material and Methods (Lines 147 -148), as well as clarified in each legend of the figures.
Point 6: Another major problem is that the total lipid amount in the cells, and the determination of the possible existence of lipidosis, relies entirely in the quantification of the fluorescence of a lipid sensitive probe. This needs to be validated by determination of true total lipid content. The fluorescence of the probe could be highly affected by the presence of the different drugs, without meaning that a real difference in lipid amount does exist.
Response 6: This raises an interesting point. Our previous publications include an article where the high content image analysis technique with fluorescent probe was conducted alongside mass spec imaging for ex vivo rat alveolar macrophages exposed to amiodarone which indicated the technique did reflect true total lipid content (see ref. 26). This paper has been added to the discussion for clarity and confirmation of the validity of the technique (Lines 294 – 298). Whilst phospholipid content with the other drugs has not been assessed, these are not known to establish a phospholipidosis phenotype in cells and the data in this article reflects that.
Point 7: To what extent the term “vacuoles” used here also includes lipid droplets, or endosomes/lysosomes?
Response 7: The term vacuoles in this paper refers to morphometric feature of a cell. Therefore, it includes all of above: lipid droplets, endosomes/lysosomes. It has been clarified in Lines 250-252.
Point 8: Drugs as amiodarone are retained into acidic compartments as a consequence of their pH-dependent membrane permeabilizing properties, which could be pertinent to define why the drug is affecting vacuolar morphology. In the same line, the authors may like to take a look to the paper by Haller at al (BBA 1860: 1152, 2018), where the effect of amiodarone and other drug have been studied on lipid containing organelles of other cells.
Response 8: The review makes an interesting point regarding the fate of amiodarone in cells. However, the scope of this paper is not to explore the mechanistic understanding of vacuole formulation but to evaluate with the high content image analysis technique can differentiate between different types of induced ‘foamy’ macrophage responses – amiodarone being an established inducer of this particular morphological feature. However, the reviewer raises an interesting point and we have included a sentence in the discussion section to cite this reference as suggested (Line 321-323).
Minor
Point 1: Please, clarify what is “lymphocyte degradation” (line 45).
Response 1: It should be “lymphocyte degeneration”. Amended (Line 47).
Point 2: Please, revise the references to figures because some of them are wrongly
mentioned.
Response 2: Amended as advised
Point 3: Extent (line 226)
Response 3: Amended as advised
Point 4: Instead of “long term”, perhaps should be better to say “medium-long term” (line
258).
Response 4: Amended as advised (Line 276).
Round 2
Reviewer 1 Report
No additional comments.
Reviewer 2 Report
The revised version of the manuscript has now addressed properly, in my opinion, the questions raised.